# Morphology and Ultrastructure of the Female Reproductive Apparatus of an Asexual Strain of the Endoparasitoid *Meteorus pulchricornis* (Wesmael) (Hymenoptera, Braconidae)

**DOI:** 10.3390/biology12050713

**Published:** 2023-05-13

**Authors:** Yusi Chen, Pengzhan Wang, Xiaohan Shu, Zhizhi Wang, Xuexin Chen

**Affiliations:** 1Hainan Institute, Zhejiang University, Sanya 572025, China; 12116095@zju.edu.cn (Y.C.); xxchen@zju.edu.cn (X.C.); 2Guangdong Laboratory for Lingnan Modern Agriculture, Guangzhou 510642, China; 3State Key Lab of Rice Biology, Ministry of Agriculture Key Lab of Molecular Biology of Crop Pathogens and Insects, and Zhejiang Provincial Key Laboratory of Biology of Crop Pathogens and Insects, Zhejiang University, Hangzhou 310058, China; 4Institute of Insect Sciences, College of Agriculture and Biotechnology, Zhejiang University, Hangzhou 310058, China

**Keywords:** female reproductive apparatus, *Meteorus pulchricornis*, morphology, parasitoids, ultrastructure, vonosomes

## Abstract

**Simple Summary:**

Female reproductive organs in the asexual strain of the endoparasitoid *Meteorus pulchricornis* were not systematically studied so far. The study aims to characterize the female reproductive system and to identify and verify parasitic factors by light and transmission electron microscopy. We present new data on the morphology and ultrastructure of the entire female reproductive system in a thelytokous strain of *M. pulchricornis*. Knowing the structure of the female reproductive system and the physiological functions of individual organs will improve our understanding of parasitoid wasp reproductive processes and biology. These findings provide a theoretical basis for further studies on host modulation and biological control.

**Abstract:**

*Meteorus pulchricornis* (Wesmael) is a solitary endoparasitoid of lepidopteran pests and a good candidate for the control of *Spodoptera frugiperda*. To elucidate the structure of the female reproductive apparatus, which may play a role in facilitating successful parasitism, we presented the description of the morphology and ultrastructure of the whole female reproductive system in a thelytokous strain of *M. pulchricornis*. Its reproductive system includes a pair of ovaries without specialized ovarian tissues, a branched venom gland, a venom reservoir, and a single Dufour gland. Each ovariole contains follicles and oocytes at different stages of maturation. A fibrous layer, possibly an egg surface protector, coats the surface of mature eggs. The venom gland consists of secretory units (including secretory cells and ducts) with abundant mitochondria, vesicles and end apparatuses in the cytoplasm, and a lumen. The venom reservoir is comprised of a muscular sheath, epidermal cells with few end apparatuses and mitochondria, and a large lumen. Furthermore, venosomes are produced by secretory cells and delivered into the lumen via the ducts. As a result, myriad venosomes are observed in the venom gland filaments and the venom reservoir, suggesting that they may function as a parasitic factor and have important roles in effective parasitism.

## 1. Introduction

Parasitic wasps depend on various and exquisite strategies to manipulate the host’s physiological conditions and render them more conducive to the development of their offspring [1,2,3]. These strategies encompass the generation of parasitic factors derived from the female reproductive system [4], or specialized embryonic cells, known as teratocytes [5]. The maternal factors, including venom, polydnaviruses (PDVs), virus-like filaments (VLFs), virus-like particles (VLPs), fibrous layers, and ovarian proteins (OP), perform a vital function for the survival of their offspring [6,7,8,9,10,11,12,13].

The female reproductive system of parasitic wasps generally consists of a pair of ovaries, oviducts, a receptaculum seminis, a venom gland, a venom reservoir, and a Dufour gland. However, the presence and structural characteristics of each organ of the female reproductive system vary among parasitoid groups or species [14]. Notably, in some endoparasitoids of Ichneumonidae and Braconidae, the ovary is enlarged to form a calyx at the junction with the oviduct. PDVs are unique symbiotic viruses and were only observed in the calyx of these two families [15,16]. In addition, the venom apparatus of parasitoid wasps is morphologically diverse, and the general morphology and ultrastructure of the venom apparatus was extensively researched, especially in the ichneumonid and braconid wasps [17,18,19,20,21,22]. The VLPs are discovered in a specialized ovarian tissue or venom apparatus of wasps in Braconidae, Ichneumonidae, Platygastridae, and Cynipidae using electron transmission microscopy. For example, VLPs in *Venturia canescens* and *Microctonus aethiopoides* are synthesized in the nucleus of the calyx and then released to the lumen of the oviduct, which is the same as PDV but devoid of DNA or RNA [23,24]. VLPs were also found in the venom gland of all three species of *Drosophila* parasitoids, *Leptopilina boulardi*, *L. heterotoma,* and *L. victoria* [25,26,27]. In *Opius caricivorae*, VLPs were found in both the epidermal cells of the ovaries and the secretory cells of the venom gland [28]. Furthermore, two categories of VLPs were classified: one resembles viruses and the other is of vesicular origin [29]. Recent studies showed that virus-resembling VLPs usually contain viral-derived genes, such as those in *V. canescens* [30] and *Fopius arisanus* [31], while some VLPs lack any known viral-associated proteins, such as in *L. heterotoma* [32]. Furthermore, the size of the VLPs varied among different parasitoid species, as did the location of VLP production within the same species. In addition to the above parasitic factors, many other viruses are also present in the female reproductive system [33,34,35]. Thus, research on the structure of the female reproductive system not only provides a basis for the research on the tissue localization of various parasitoid factors, but also has important implications for the study of the physiology of parasitoid wasps.

*Meteorus pulchricornis* (Wesmael) (Hymenoptera, Ichneumonoidea, and Braconidae) (Figure 1) is a highly polyphagous larval parasitoid of a wide range of lepidopteran species, including *Spodoptera frugiperda*, a worldwide pest that causes significant economic losses by attacking and destroying crops [36,37,38]. Though it is described that single-membraned VLPs are filled within the lumen of the venom gland of *M. pulchricornis*, a recent omic data analysis shows that no viral protein is present in these VLPs [39,40,41,42]. Thus, we substitute the term “venosomes” (venom vesicles) for VLPs as recommended [29]. Interestingly, both sexual and asexual strains of *M. pulchricornis* are discovered in nature. Sexual strain is discovered in Europe and Japan and thelytokous strains can be found in East Asia, Australia, and New Zealand [25,38,43,44,45,46,47]. These two strains raise the question of whether they use the same reproductive strategies. However, despite the venosomes, there are no reports on the morphology and ultrastructure of the complete female reproductive system of *M. pulchricornis*.

In this study, we focused on the whole reproductive system of an asexual strain of *M. pulchricornis* from China, as well as the presence of possible parasitic factors in this species. We examined the morphology of the female reproductive system and its ultrastructure using light and transmission electron microscopy (TEM). Interestingly, we observed that the venosomes in the venom gland filaments could be stored in vesicles, delivered via duct cells, and eventually released into the lumen. In addition, we did not find any evidence of viruses or virogenic stroma in the ovarian tissue. These results provide further insight into the reproductive biology and venosomes conveying of *M. pulchricornis*, which could have important implications for the use of this species as a biocontrol agent in the future.

## 2. Materials and Methods

### 2.1. Insect Rearing and Parasitization

*S. frugiperda* and the asexual strain of *M. pulchricornis* were collected in October 2020 from corn fields in Ningbo (N 29°52′, E 121°31′), Zhejiang Province (China). *S. frugiperda* larvae were fed with artificial diets and reared under laboratory conditions of 26 ± 1 °C, 65 ± 5% RH, and a 14 h light and10 h dark photoperiod. For parasitization, a third instar *S. frugiperda* larva was exposed to a female *M. pulchricornis* in a transparent plastic tube (2.2 cm diameter, 8 cm height). After the female wasp was released into the transparent plastic tube, parasitism behaviour was observed. When the female wasp stung the host body with its ovipositor for a few seconds, we collected the *S. frugiperda* larvae and reared them individually in petri dishes (6 cm diameter). Adult female wasps were reared in plastic containers (10 cm diameter, 8.5 cm height), with 10% honey solution supplied daily via cotton. Thirty live adult females were selected for dissection.

### 2.2. Dissection of Female Reproductive Apparatus and Its Light Microscopy

Parasitoid female adults were anaesthetized on ice, and then we dissected the female reproductive apparatus under a Leica S9E microscope (Leica Microsystems, Wetzlar, Germany) in phosphate buffer (0.1 M, pH 7.2). The morphology was observed and photographed with a Keyence VHX-6000 super depth-of-field microscope (Keyence Corporation, Osaka, Japan).

### 2.3. Transmission Electron Microscopy

The ultrastructure of the female reproductive apparatus was studied. Different parts of the system, such as the ovaries (divided into three parts: germarium, vitellarium, and the rest, including the pedicel, respectively), venom glands, venom reservoirs, and Dufour glands (Figure 2B–F) were treated using the following procedures: (1) fixed overnight at 4 °C in 2.5% glutaraldehyde; (2) rinsed three times in phosphate buffer (0.1 M, pH 7.0) for 15 min each time; (3) double fixation with 1% osmium acid solution for 1.5 h; (4) carefully removed the osmium acid waste solution and rinsed with phosphate buffer (0.1 M, pH 7.0) three times for 15 min each time; (5) dehydrated with ethanol solutions of gradient concentrations (including 30%, 50%, 70%, 80%, 90%, 95%, and 100%) for 15 min at each step, and then transferred to pure acetone for 20 min; (6) infiltration with a mixture of Spurr embedding agent and acetone at a ratio of 1:1 and 3:1 for 1 h and 3 h, respectively; (7) infiltration with a pure embedding agent at room temperature overnight; and (8) placed in Eppendorf containing Spurr resin and heated at 70 °C for 24 h, and then sliced in Leica Em Uc7 ultratome (Leica Microsystems, Wetzlar, Germany). The ultrathin sections were stained with uranyl acetate and alkaline lead citrate and observed in a Hitachi H-7650 (Hitachi, Tokyo, Japan) transmission electron microscope.

## 3. Results

### 3.1. Morphology of the Female Reproductive Apparatus

The female reproductive system consists of a pair of ovaries, a venom reservoir, a venom gland, and a Dufour gland (Figure 2A). Each ovary consists of 7–8 yellowish ovarioles, within each of which the oocytes are arranged in a single row, with slender terminal filaments protruding from the anterior end of the ovarioles and clustered into bundles. Each ovariole contains five to six eggs at different stages of maturation (Figure 2B). The mature eggs in the basal region of ovarioles are elongated with an egg stalk, which is involved in the attachment of the egg to the host’s peripheral tissue (Figure 2C). The venom glands are two branched tubular-shaped whitish filaments, with one end closed and free and the other end converging at the base to form a common venom gland secretory duct connecting to the end of the venom reservoir. The venom reservoir is oval and opaque under the light microscope (Figure 2D). The Dufour gland, located at the base of the duct and closely associated with the venom reservoir, is a simple, unbranched, translucent tubular gland (Figure 2E). The schematic drawing of the whole female reproductive system shows the organization of each section. The sting is usually enclosed in a sheath lobe. The duct of the venom gland opens at the base of the venom reservoir. The venom reservoir ends at the beginning of the oviduct, together connected to the Dufour gland. The ovary is connected to the median oviduct by a pair of lateral oviducts, which eventually connect to the above parts (Figure 2F).

### 3.2. Ultrastructure of Ovary

During the ongoing stages of oogenesis, each ovariole contains oocytes of different developmental stages (Figure 3). In the germarium (Figure 3A), many undifferentiated oogonia are closely arranged. The oogonia are round, with a round and large nucleus that occupies almost the entire cell and is rich in chromatin. Around the nucleus, there are many spherical or rod-shaped mitochondria, which provide energy for later cell division. The ovarian sheath is thin and consists of only a flattened layer of epithelial cells (Figure 3A,B). The trophocytes embed plenty of mitochondria and are closely arranged (Figure 3C). During oocyte development, the oocytes increase in size, and lipid droplets and yolk proteins accumulate (Figure 3D). There are microvilli structures between oocytes and follicle cells, which may be constructed for communications (Figure 3E). In the course of chorion formation, the follicle cells degenerate and gradually become flattened. When vitelline accumulation is almost complete, the vitelline membrane and chorion begin to form (Figure 3F,G). Finally, the egg is encircled by a large number of lipid droplets and yolk granules, and surrounded by a vitelline membrane and chorion. The chorion is divided into the endochorion and the exochorion, which is covered by a fibrous layer (Figure 3G). Later in the development of the ovary, the mature egg is discharged into the oviduct (Figure 3H). The oviduct is covered by an ovarian sheath and lined with a thin epithelial layer, both of which together form the border of the lumen (Figure 3I).

### 3.3. Ultrastructure of Venom Gland

Transversal sections show that secretory units, including secretory cells and ducts are arranged around the lumen of the venom gland (Figure 4A). Secretory cells are characterized by clusters of secretory granules, rough endoplasmic reticulum, mitochondria, vesicles, and end apparatus, which show intense signs of protein synthesis. The nucleus is spherical. The cytoplasm is densely packed with rough endoplasmic reticulum arranged longitudinally in a tight parallel or tubular pattern. The number of vesicles in the cytoplasm is numerous and varies in size. The lumen of the venom gland is filled with venosomes (Figure 4A). The nucleus of the intima layer is close to the lumen, which is filled with venosomes (Figure 4B). The end apparatus is very large and internal microvilli are radially distributed around the center. The presence of microvilli greatly increases the surface area for the venosome secretion and is connected to the lumen of the venom gland by a duct (Figure 4C,D). Venosome precursors, the structure of which is different from that in the end apparatus and lumen, are stored in cytoplasm vesicles (Figure 4E). The Golgi apparatus and mitochondria are distributed around vesicles, indicating the possibility of the active transportation of materials (Figure 4F).

### 3.4. Ultrastructure of Venom Reservoir

The venom reservoir consists of an external muscle layer, an internal layer of epidermal cells, and a lumen (Figure 5A). A large number of mitochondria and muscle fibres surround the epithelial cells (Figure 5A,B). This structure may help to drain the venom out of the reservoir by extrusion. The epithelial cells have an elongated nucleus that occupies almost the entire cell (Figure 5A). At the periphery of the nucleus, in addition to several mitochondria, there are a few end apparatuses in the cytoplasm that also occur in the venom gland (Figure 5B), suggesting that the venom reservoir of *M. pulchricornis* not only has a storage function, but also secretes a few venosomes. However, in comparison with the end apparatus of the venom gland, its central canal of the reservoir is smaller and has more intensive microvilli. The lumen of the venom reservoir is lined with a thin, uniform inner membrane (Figure 5B), and is centrally occupied with plenty of venosomes, which are single-membrane vesicles, measuring 207 ± 20 nm in diameter (*n* = 25) and composed of dense or semi-dense materials (Figure 5C).

### 3.5. Ultrastructure of Dufour gland

The Dufour gland has a relatively large lumen surrounded by a single layer of very thin epithelial cells (Figure 6A). These epithelial cells are composed of secretory granules and mitochondria. Moreover, the cells have many vacuolar structures and muscles, suggesting plenty of compounds are secreted into the lumen and pumped into the host during oviposition (Figure 6B,C). There are also a few electronlucid secretory granules with no distinct membrane.

## 4. Discussion

The female reproductive system of *M. pulchricornis* includes two ovaries, a branched venom gland, a large venom reservoir, and a Dufour gland, which corresponds to the general organization described in other parasitic wasps [14,48].

In our study, the oogenesis of *M. pulchricornis* conforms to five stages, consistent with previous studies of *N. vitripennis* [14]. During choriogenesis, the follicle epithelium degenerates gradually and forms the chorion. Additionally, Barratt et al. [24] and Wan et al. [28] found VLPs in the ovarian epithelial cells of *M. aethiopoides* and *O. caricivorae*, respectively. We examined the follicular epithelium at each developmental stage of *M. pulchricornis* and did not discover any similar materials. In addition, PDVs are formed exclusively in the calyx of the ovary [49,50]. In *M. pulchricornis*, no distinct enlarged calyx region (Figure 1B) and no typical virus particles (Figure 2H,I) were observed at the base of the ovary, suggesting no PDV was present in this species. Despite the well-described PDVs, the presence of other viruses in the ovaries remains questionable due to the biodiversity of the virus. Analysis by light microscopy and TEM alone cannot rule out the absence of all kinds of viruses. The existence of viruses in the reproductive system will be studied by integrating more data in the future.

The surface of mature eggs covers a fibrous layer, and this structure is quite similar to that described in *Trichomalopsis shirakii* [22] and *Cotesia vestalis* [51]. Some parasitoids may have a coating that prevents their recognition by or adhesion to the host’s hemocytes [52]. The fibrous layer of eggs seems to be an egg surface protector for passive evasion of host immunity [52,53,54]. Therefore, we speculated that the fibrous layer in *M. pulchricornis* might act as a parasitic factor to escape the host immune response.

The venom apparatus of *M. pulchricornis* consists of a venom gland and a venom reservoir with a Dufour gland. The venom gland and Dufour gland are similar to those described in Braconidae species, such as *Cardiochiles nigriceps* [55], *Habrobracon hebetor* [56], *Bracon cephi,* and *B. lissogaster* [57]. It is well known that the general morphology of the venom apparatus is very diverse among Hymenoptera, making it a significant feature for inferring the evolutionary relationships among parasitic wasps [58]. The ultrastructure of the secretory cells of the venom gland is also quite similar to those described in other parasitoids, such as *Diadromus collaris* [59] and *T. shirakii* [22]. They all belong to the class III gland cells, with a cuticular duct penetrating the gland cell [60], but the venoms of the latter two species do not contain any venosomes.

Edson and Vinson classified the venom apparatus of braconid wasps into two types: Type I venom reservoirs are surrounded by muscle and have relatively thick membranes with an ancestral trait, which is a lower evolutionary model; Type II venom reservoirs have less muscle and thin inner membranes, which is a higher evolutionary model [19]. The venom reservoir of *M. pulchricornis* belongs to type II. Moreover, the degradation of the muscles and the epidermal cell of the venom reservoir probably makes room for venom storage. In addition, the epidermal cell of the venom reservoir has several mitochondria and end apparatuses, suggesting these cells are probably involved in venom protein secretion and venosome delivery. The structure is similar to that in *B. mellitor* and *B. hebetor*, but different from *C. nigriceps* and *Pteromalus puparum*, whose venom reservoir only serves as a venom storage organ [19,28,61].

As a major parasitic factor, venom seems to play an important role in parasitism [6,62,63,64]. Their constituents vary between species. In addition to the liquid venom, several species, including *M. leviventris* [19] and *L. victoria* [25], own massive VLPs or venosomes in the venom gland. In *M. pulchricornis*, venosomes, devoid of nucleic acids, are formed in the secretory cells of the venom glands and appear as vesicular particles, which were previously described as single-membrane vesicles and filled or semi-filled with electron-dense materials [39]. The venosomes in our study are structurally consistent with the previously described ones [39], but the size is larger, i.e., the average venosome diameter of this study is about 210 nm, while the previously described ones averaged 180 in the major axis, possibly due to the geographic differential between these two strains. However, unlike what is known about PDVs produced in wasp ovaries, up to now, the exact mechanism of the biogenesis of venosomes, including *Meteorus* spp. [19,42], remains to be investigated.

The ultrastructure of the Dufour gland was well studied in many Hymenoptera, including ants, bees, and several families of parasitic wasps [17,65,66]. The ultrastructure of the epithelial cells in *M. pulchricornis* is similar to those mentioned above, but the structures are simpler. From the ultrastructural study of the Dufour gland in *M. pulchricornis*, the epithelial cell possesses many vacuole structures and several secretory granules. Dufour gland secretions consist of a remarkable diversity of compounds and act as a lubricant at the time of oviposition or as a parasitism-marking pheromone for host discrimination in braconid and ichneumonid wasps [55,67,68,69,70,71,72]. With myriad compounds in the Dufour gland of *M. pulchricornis*, we speculated a diversity function of this organ including acting as a lubricant or a parasitism-marking pheromone.

In summary, our findings indicate that a fibrous layer adjoins the mature egg surface, and numerous venosomes exist within the venom glands and venom reservoir. These particles are generated within the venom glands, stored in vesicles, and conveyed to the lumen of the venom gland by the ducts. Though they play an important role in parasitism [39,40], the origin and biogenesis of the venosomes are still not yet known. As such, further analysis is required to ascertain the exact role played by the fibrous layer and venosomes discovered in our study, particularly concerning the host–parasitoid relationship.

## 5. Conclusions

Our study reported the morphology and ultrastructure characteristics of the reproductive system of a thelytokous parasitic wasp *M. pulchricornis* (Wesmael) from China. These results provide direct morphological evidence that the venosomes in the venom apparatus of *M. pulchricornis* and the fibrous layer attached to the mature egg surface may serve as parasitic factors to manipulate the host.

## Figures and Tables

**Figure 1 biology-12-00713-f001:**
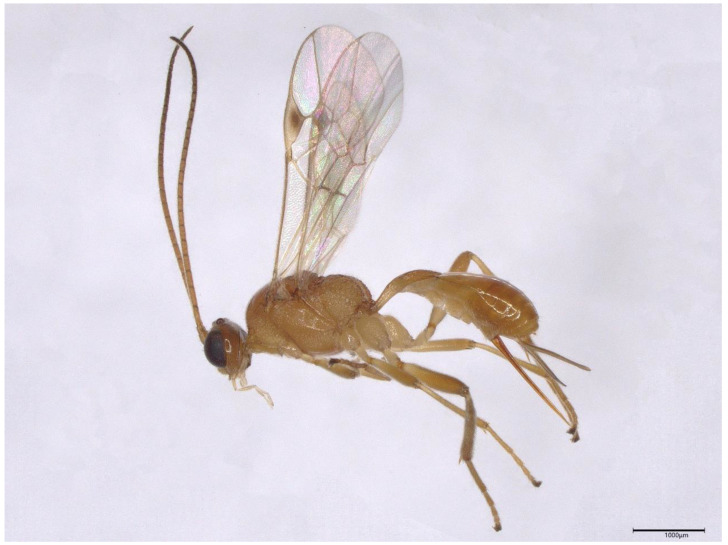
Side view of *M. pulchricornis* (female). Scale bars: 1000 μm.

**Figure 2 biology-12-00713-f002:**
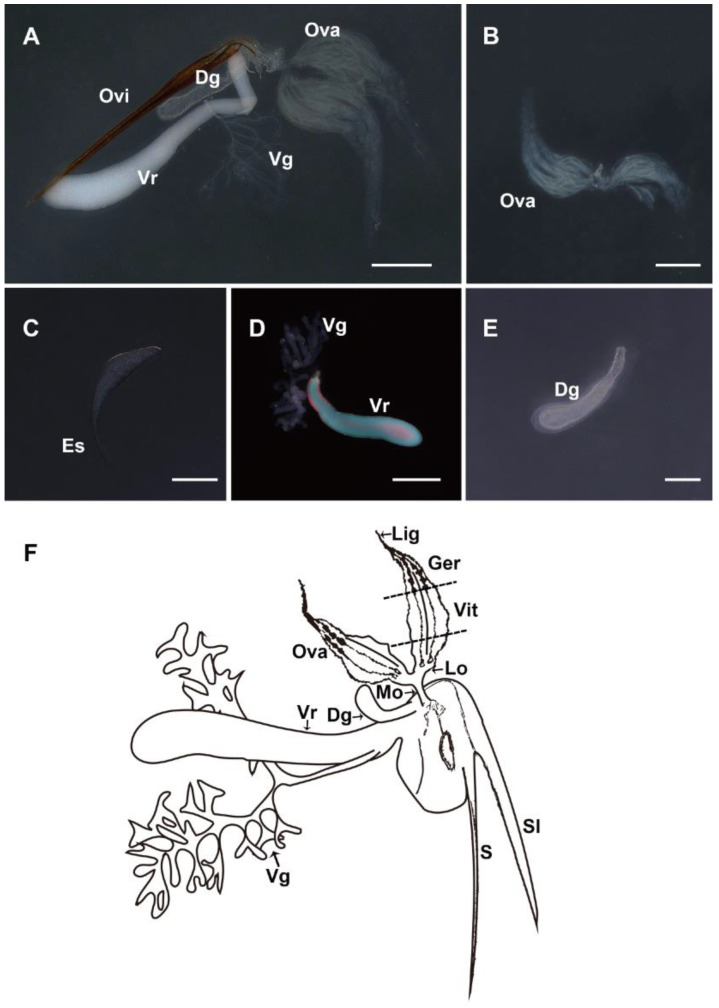
Morphology of the female reproductive apparatus of *M. pulchricornis*. (**A**) General morphology of the whole system. (**B**) Morphology of the ovary. (**C**) Morphology of the mature egg with egg shea. (**D**) Magnification of the venom apparatus. The venom gland has two branches, and the venom reservoir is filled with a cloudy fluid. (**E**) Magnification of the Dufour gland. (**F**) Schematic drawing of the organization of the whole female reproductive system. Ovi, ovipositor; Ova, ovary; Vr, venom reservoir; Vg, venom gland; Dg, Dufour gland; Es, egg stalk; S, sting; Sl, sheath lobe of the sting; Mo, median oviduct; Lo, lateral oviduct; Vit, vitellarium; Ger, germarium; and Lig, ligament. Scale bars: (**A**,**B**,**D**) = 500 µm, (**C**) = 100 µm, and (**E**) = 250 µm.

**Figure 3 biology-12-00713-f003:**
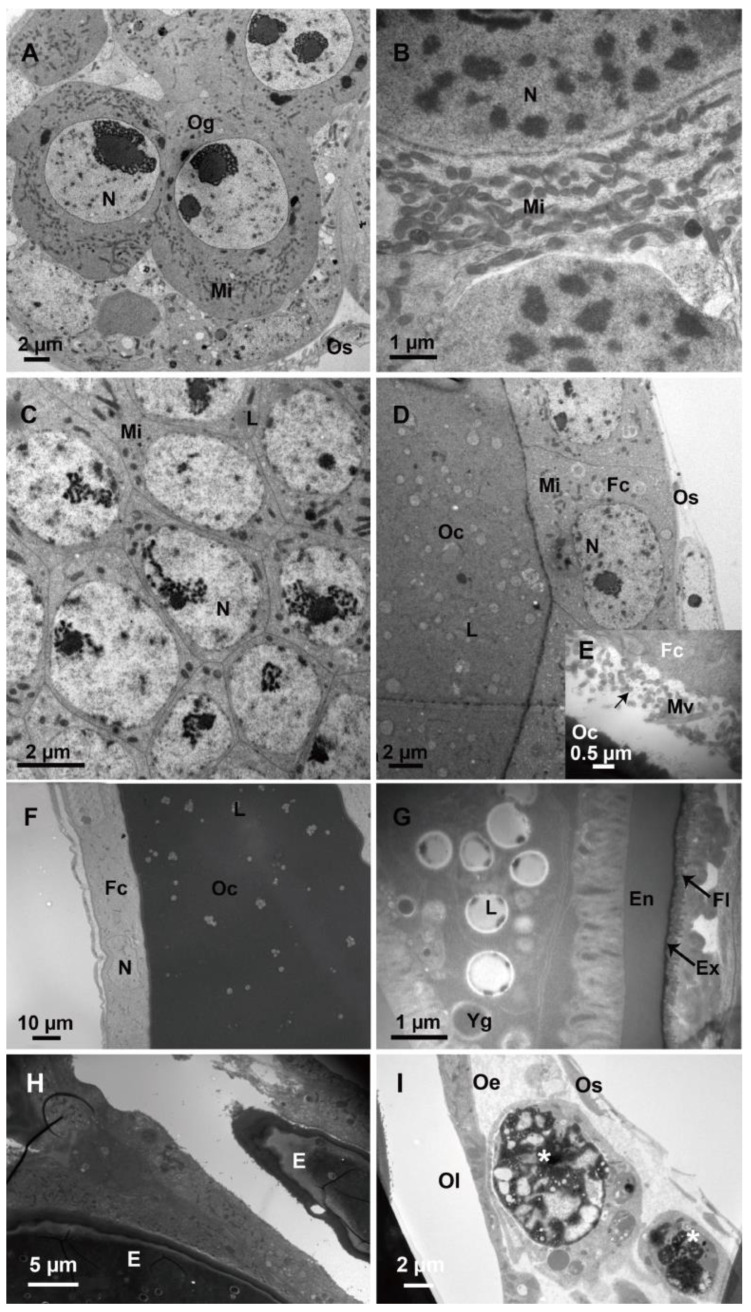
Ultrastructure of different developmental stages of oocytes in the ovary. (**A**,**B**) Undifferentiated oogonia in the germarium at the end of the ovarioles. (**C**) Tightly packed trophoblast cells in the germarium. (**D**) Oocytes and follicle cells in the vitellarium. (**E**) Microvilli structures between oocytes and follicle cells showing material communications (arrows). (**F**) Follicle cells and oocytes during chorion formation. (**G**) Mature oocyte with a fibrous layer outside the chorion (arrows). (**H**) Two mature eggs in the oviduct. (**I**) A layer of epithelial cells (asterisk) forms the border of the oviduct lumen. Og, oogonia; Fc, follicle cell; Oc, oocyte; Os, ovariole sheath; Mv, microvilli; N, nucleus; Mi, mitochondria; L, lipid droplet; Ri, ribosome; Yg, yolk granule; En, endochorion, Ex, exochorion; Fl, fibrous layer; Oe, oviduct epithelium; and Ol, oviduct lumen, E, and egg.

**Figure 4 biology-12-00713-f004:**
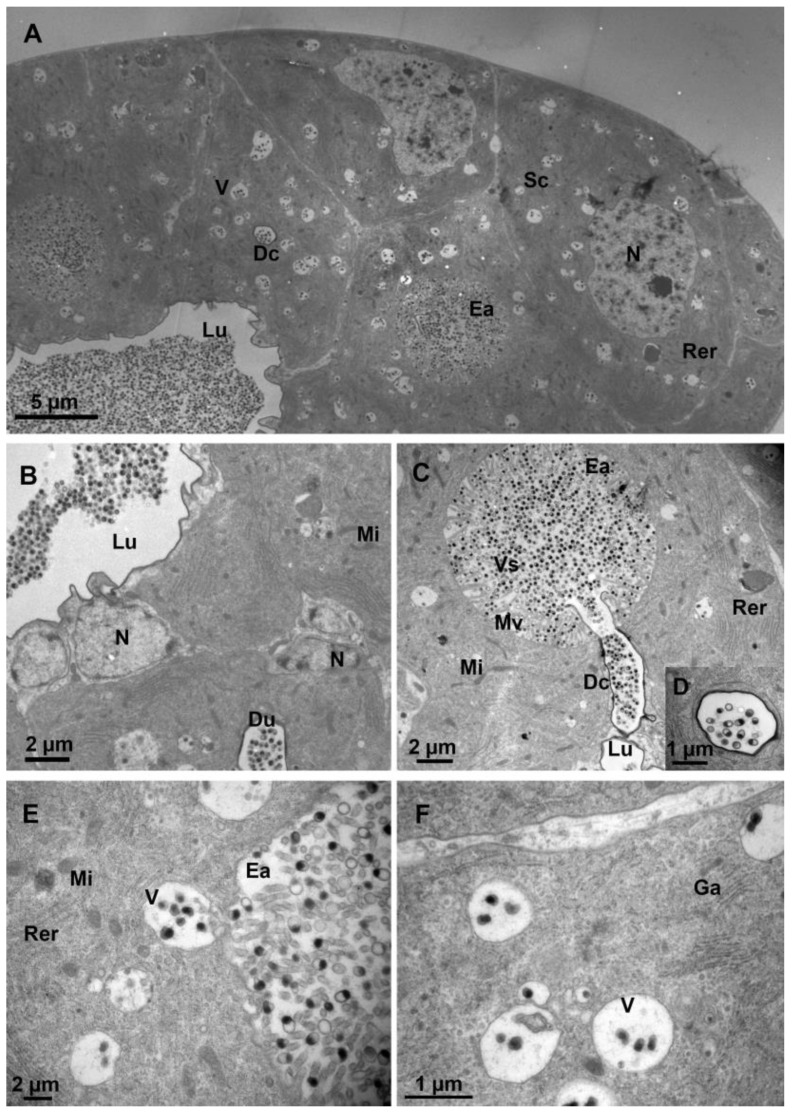
Ultrastructure of the venom gland. (**A**) Part of a cross-section of the venom gland showing the lumen filled with venosomes. (**B**) The lumen and the surrounding cells. (**C**) The end apparatus with many microvilli and venosomes. The duct connecting the end apparatus to the lumen of the venom gland. (**D**) Magnified view of the cross-section of a duct. (**E**) Vesicle storing and transporting the venosome precursors. (**F**) Vesicles with venosome precursors inside and associated with the Golgi apparatus. N, nucleus; Sc, secretory cell; V, vesicle; Lu, lumen; Mi, mitochondria; Rer, rough endoplasmic reticulum; Ea, end apparatus; Dc, duct; Mv, microvilli; Ga, Golgi apparatus; and Vs, venosomes.

**Figure 5 biology-12-00713-f005:**
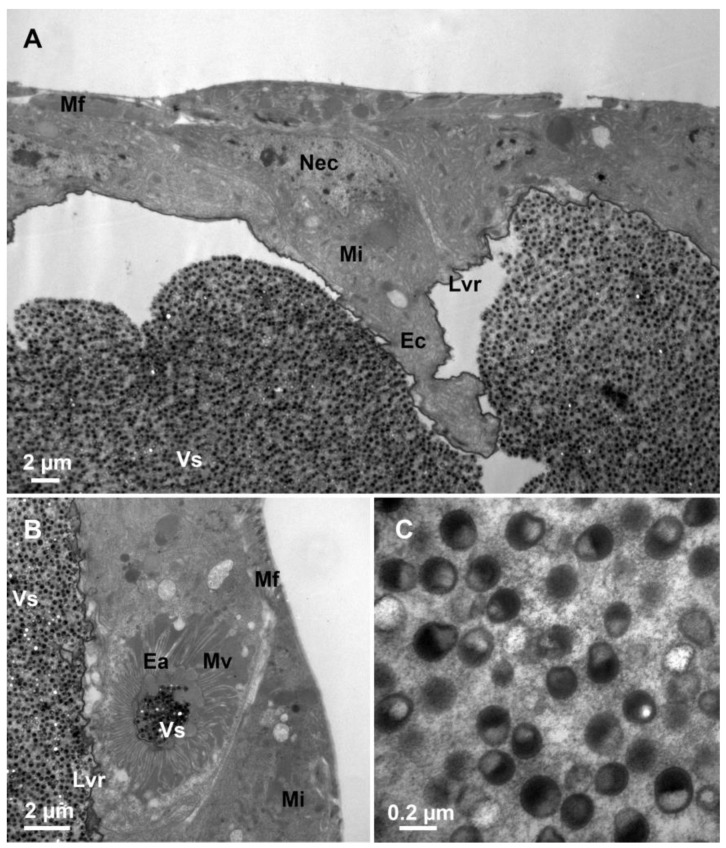
Fine structure of the venom reservoir. (**A**) The external muscle layer, the internal epidermal cell layer with an elongated nucleus, and the inner membrane layer of the venom reservoir wall. (**B**) The epidermal cell with mitochondria and end apparatus. (**C**) Lumen of the venom reservoir filled with venosomes. Mf, muscle fibre; Nec, the nucleus of the epidermal cell; Mi, mitochondria; Ec, epidermal cell; Lvr, the lumen of venom reservoir; Ea, end apparatus; Vs, venosomes; and Mv, microvilli.

**Figure 6 biology-12-00713-f006:**
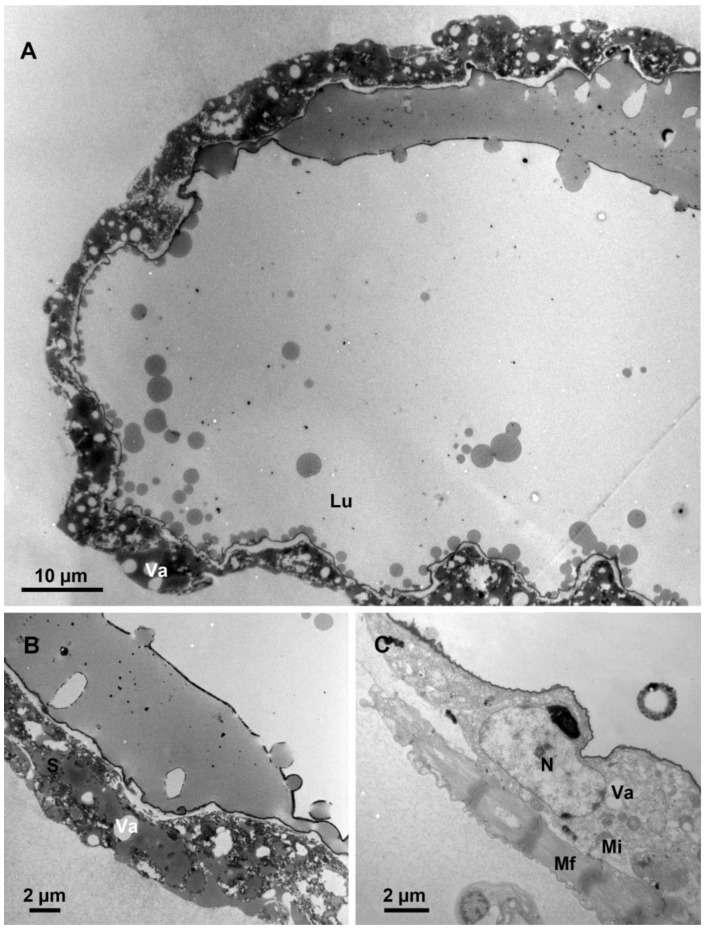
Fine structure of the Dufour gland. (**A**) Part of the Dufour gland showing a large lumen surrounded by epithelial cells. (**B**) Part of the epithelial cell showing vacuoles and secretory granules. (**C**) The detail of the large nucleus in an epithelial cell and the vacuoles. Lu, lumen; Mi, mitochondria; N, nucleus; Mf, muscle fibre; S, secretory granules; and Va, vacuole.

## Data Availability

Data are contained within the article.

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
