# Peer review of "Morphology and Ultrastructure of the Female Reproductive Apparatus of an Asexual Strain of the Endoparasitoid Meteorus pulchricornis (Wesmael) (Hymenoptera, Braconidae)"

_biology, 2023, doi:10.3390/biology12050713_

Round 1

Reviewer 1 Report

The paper is about the morphology and ultrastructure of female reproductive system of the parasitoid wasp and does not offer much novelty. Although there was some indication of the parasitic factor study but the study could not put forth any worthwhile finding in that regard. There is a detailed paper already in place by Gatti et al. (2021) entitled Proteo-Trancriptomic Analyses Reveal a Large Expansion of Metalloprotease-Like Proteins in Atypical Venom Vesicles of the Wasp Meteorus pulchricornis (Braconidae).

The figures LM and TEM are good but the corresponding alphabets need to be of small size.

The queries are marked in the text

93: Usually the viruses or virogenic material is being reported from ovarian tissue but in the present study how did the authors rule out their presence. Was it only based on their presence in the TEM sections?

208: In fig 5, why is the internal epidermal cell layer degenerated?

279: How did the authors know that the secreted products are mainly lipids and may be be lubricants or pheromones? Please give justification.

The authors talk about secretory activity but did not mention golgi complex at any point although they wrote about vesicles.

Above all, the merit of the MS lies in its comparison with other braconids to assess the life cycle and parasitic adaptations.

Language needs a thorough revision.

Reviewer 2 Report

The paper of Yu-Si Chen et al deals with a morphological study carried out on the ultrastructural organization of the female reproductive system in a parasitic wasp. Generally speaking, the paper is of average quality, the pictures are good and include several ultrastructural details of the described structures. However I have comments that need the author’s attention.

I recommend changing the font used in the figures. Use a Sans Serif font (arial would work well, but also others) to do not make the graphics heavy.

Other comments in the annotated PDF.

English is average, but a general check is needed, The most of the criticism were found kin the discussion (see annotated PDF).

Round 2

Reviewer 2 Report

The authors made the changes according to the comments, therefore the paper can be accepted for publication